# AN EFFICIENT MULTI-TASK TRANSFORMER FOR 3D HEAD ALIGNMENT

## ABSTRACT

In the research of 3D head alignment, few prior works focus on information exchange among different vertices or 3DMM parameters in regression. On the other hand, there is a drawback that using high-resolution feature maps makes algorithms memory-consuming and not efficient. To solve these issues, we first propose a multi-task model equipped with two transformer-based branches which further enhances the information communication among different elements through self-attention and cross-attention mechanisms. To solve the problem of low efficiency of high-resolution feature maps and improve the accuracy of facial landmark detection, a lightweight module named query-aware memory (QAMem) is designed to enhance the discriminative ability of queries on low-resolution feature maps by assigning separate memory values to each query rather than a shared one. With the help of QAMem, our model is efficient because of removing the dependence on high-resolution feature maps and is still able to obtain superior accuracy. To further improve the robustness of the predicted landmarks, we introduce a multi-layer additive residual regression (MARR) module that can provide a more stable and reliable reference based on the average face model. Furthermore, the multi-information loss function with Euler Angles Loss is proposed to supervise the network with more effective information, making the model more robust to handle the case of atypical head poses. Extensive experiments on two public benchmarks show that our approach can achieve state-of-the-art performance. Besides, visualization results and ablation experiments verify the effectiveness of the proposed model.

## 1 INTRODUCTION

3D face alignment is an essential task for face-related computer vision problems, such as facial landmark detection Chandran et al. (2020), 3D head pose estimation (Murphy-Chutorian & Trivedi, 2009), face tracking (Deng et al., 2019), 3D face reconstruction (Dou et al., 2017; Feng et al., 2018), and face editing (Thies et al., 2016). These applications require the model to be accurate and robust to varied facial appearance, different age groups, atypical head poses, and even in-the-wild deployment conditions, which remains a significant challenge for existing methods.

According to the ways of generating vertices, previous works can be mainly classified into two categories: landmark coordinates regression (Feng et al., 2018; Jackson et al., 2017) and 3D Morphable Model (3DMM) parameters regression (Blanz & Vetter, 1999; Zhu et al., 2019). Landmark coordinates regression directly transforms feature maps into vertices coordinates. Since the high dimensionality of the network, these algorithms are memory-consuming in inference. With regard to another strategy, numerous works (Zhu et al., 2019; Guo et al., 2020; Wu et al., 2021; Bulat & Tzimiropoulos, 2017a) focus on regressing a set of 3DMM parameters to predict facial geometry, which is more efficient than directly regressing the coordinates of all the dense 3D vertices. The above models rely on datasets with 2D-to-3D pairing information, while the current 3D facial datasets are limited in scale and captured not-quite-in-the-wild (Sanyal et al., 2019). Recently, DAD-3DHeads (Martyniuk et al., 2022) presents a dense and diverse large-scale dataset for 3D face alignment, which is an in-the-wild dataset and covers abundant annotations of diverse attribute information. To achieve end-to-end training on the DAD-3DHeads dataset, the DAD-3DNet with a differential FLAME (Li et al., 2017) decoder is also proposed to recover the 3D head geometry by regressing the 3DMM parameters.

Despite achieving excellent performance, existing methods face three main drawbacks: (1) Due to the lack of information communication among different vertices or 3DMM parameters, the traditional convolutional neural network (CNN) limits the discriminability of predictions. (2) Using the high-resolution feature maps makes these algorithms memory-consuming and not efficient. (3) The rich annotation information of 3D coordinates is not fully exploited, which decreases the robustness of the model to atypical scenes.

In this work, a multi-task 3D head alignment framework based on the transformer is proposed to overcome the first drawback, where the 2D facial landmark detection task and the 3DMM parameters prediction task are paralleled in the form of two transformer branches. With an auxiliary task of 2D facial landmark detection, the performance of 3D face alignment is effectively improved. Besides, our model is also the first work to regress 3DMM parameters through Transformers, where the cross-attention mechanism effectively enhances the information communication among task-oriented queries and extracted feature maps in the designed decoder. To deal with the second drawback and further improve the accuracy with minimum computational burden, we propose a lightweight module named query-aware memory (QAMem), which makes up the accuracy loss from lower feature map resolutions. To enhance the robustness of the predicted landmarks, we calculate the average vertices coordinates of the training set, then a multi-layer additive residual regression (MARR) module is designed in the decoder to guide the detection under the reference of an average face model. To tackle the third drawback, the multi-information loss function is used to optimize the network. The loss function of baseline consists of three components including Landmark Regression Loss, 3D Head Shape Loss, and Reprojection Loss. To enhance the predictive ability in the case of atypical head poses, we introduce the Euler Angles Loss to provide further information supervision for network optimization.

Our contributions can be summarized as follows:

- A Transformer-based multi-task framework is proposed for 3D head alignment, where the performance of 3D face alignment is effectively improved with the help of multi-task structure. This is the first work to regress 3DMM parameters through Transformers, where the cross-attention mechanism is effective to achieve the information communication among different elements.
- A novel QAMem module is proposed to improve the accuracy so that high-resolution feature maps are no longer necessary for obtaining superior accuracy.
- A module named MARR is designed in the decoder to improve the robustness of the predicted landmarks by providing reference based on the average face model.
- We further introduce the Euler Angles Loss to the multi-information loss function for network optimization, which enhances the predictive ability in the case of atypical head poses.

Finally, a multi-task model with multi-information supervision is proposed and named Trans3DHead that is efficient for 3D head alignment.

## 2 RELATED WORK

### 2.1 2D FACIAL LANDMARK DETECTION

For 2D facial landmark detection task, previous works are mainly divided into two categories including coordinate regression-based and heatmap-based methods.

Coordinate regression-based methods focus on directly regressing the landmark coordinates and usually apply the cascaded structures for accurate locations. A drawback of them is that the landmark coordinates are regressed by a fully connected output layer, so these methods usually ignore the spatial correlations of the locations. Recently, some methods based on visual transformers have shown remarkable success. To handle complex scenarios, RePFormer (Li et al., 2022b) proposes a pyramid transformer head (PTH) and decomposes the regression of landmark coordinates into multiple steps by using multi-level information in pyramid memories to predict the residual coordinates. DTLD (Li et al., 2022a) proposes a cascaded deformable transformer based on the deformable attention in Deformable DETR (Carion et al., 2020), which improves the detection performance with a few parameters increasing.

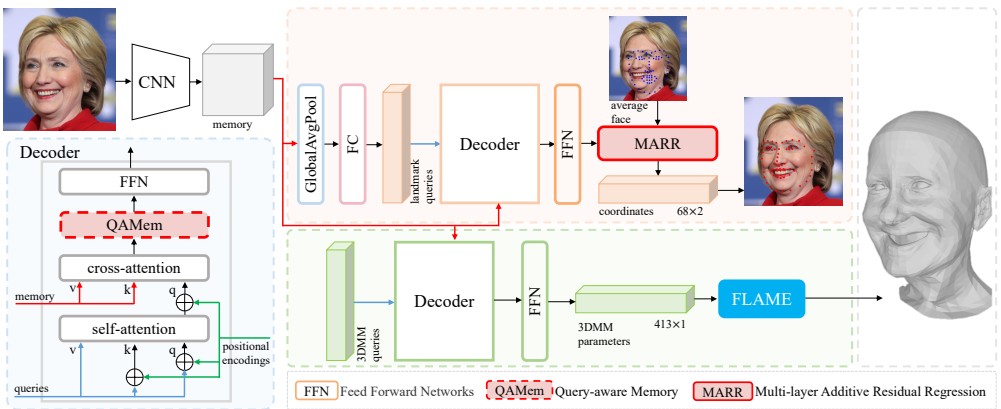

Figure 1: The overview of the proposed Trans3DHead, which is an efficient multi-task transformer for the full 3D head alignment task and can reconstruct the shape of entire face, the full 3D head and neck. Our model mainly consists of a facial landmark detection branch and a 3DMM parameters regression branch, where query-aware memory (QAMem) and multi-layer additive residual regression (MARR) modules in red color are designed only for the facial landmark detection branch. The queries, memory, and positional encodings input to the decoder are represented by blue, red, and green lines, respectively.

Heatmap-based methods (Kumar et al., 2020; Wang et al., 2019; Sun et al., 2019) rely on high-resolution feature maps to achieve precise localization which can effectively maintain the original spatial relation among pixels. LUVLi (Kumar et al., 2020) proposes an end-to-end trainable model based on U-Net (Ronneberger et al., 2015), which not only estimates landmark location but also corresponding uncertainty and visibility likelihood. Using the stacked Hourglass (HG) architecture (Newell et al., 2016) as the backbone, Awing (Wang et al., 2019) analyses the loss function for heatmap and proposes the Adaptive Wing loss to adaptively penalize loss. Despite the superior performance, these methods have high computational costs because of the calculation on high-resolution feature maps. To address this issue, PIP (Jin et al., 2021) uses low-resolution feature maps to predict heatmap and offset simultaneously, which largely reduces inference time.

Different from the existing works relying on labels of 2D facial landmarks, we aim to further explore a comprehensive model that not only meets the demand of 2D facial landmark detection but also effectively achieves 3D face alignment tasks by making full of the 3D ground truth.

## 2.2 3D FACE ALIGNMENT

3D face alignment is concerned about fitting a face model to an image. Instead of regressing 2D landmarks from the facial image, 3DDFA (Zhu et al., 2019) directly considers face alignment as a 3DMM fitting task, utilizing a cascaded CNN as the regressor. 3DDFA-V2 (Guo et al., 2020) improves 3DDFA to balance accuracy and speed. It exploits a more lightweight backbone like MobileNet (Howard et al., 2017) and further proposes meta-joint optimization to dynamically optimize 3DMM parameters. RingNet (Sanyal et al., 2019) uses FLAME as a decoder to generate 3D faces without any 2D-to-3D supervision, which is an end-to-end trainable network making full use of the shape constancy. SynergyNet (Wu et al., 2021) studies the collaborative relation between 3DMM parameters and 3D landmarks, which further enhances the information flow by reversely predicting 3DMM parameters from sparse 3D landmarks. FAN (Bulat & Tzimiropoulos, 2017a) adopts a stack of four HG networks for landmark localization and uses the hierarchical, multi-scale, and parallel binary residual blocks (Bulat & Tzimiropoulos, 2017b) to replace all bottleneck blocks used in HG, achieving remarkable accuracy on both 2D and 3D face alignment. DAD-3DNet (Martyniuk et al., 2022) is a recent approach designed to regress 3DMM parameters and reconstruct the 3D head geometry using a differential FLAME decoder. It is end-to-end trainable on the provided dataset named DAD-3DHeads with rich annotations. Based on DAD-3DNet, DAD-3DNet+ (Zeng et al., 2023) leverages EG3D (Chan et al., 2022) and Neural Radiance Field (NeRF) (Mildenhall et al., 2021) to generate multi-view images to handle the lack of multi-view in-the-wild training data.

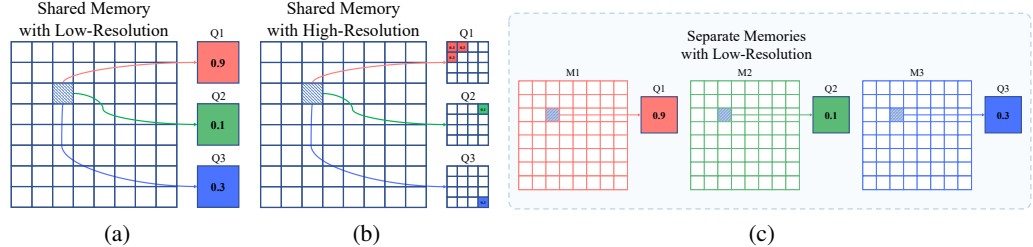

Figure 2: Different memory mechanisms. (a) A shared memory with low-resolution feature maps. (b) A shared memory with high-resolution feature maps. (c) Separate memories with low-resolution feature maps.

However, few of these approaches explore the internal relation of the 3D head geometry, such as the discriminability among predictions. In contrast, our model focuses on the information communication among different vertices or 3DMM parameters.

## 3 METHODS

### 3.1 MULTI-TASK 3D ALIGNMENT NETWORK

The overall architecture of the proposed method is illustrated in Figure 1 and the two red colored parts are newly designed modules introduced in the following sections. Our architecture consists of (i) a CNN as the backbone to extract feature maps, (ii) a auxiliary facial landmark detection branch to predict 2D landmarks, and (iii) a main 3DMM parameters regression branch to regress 3DMM parameters and reconstruct the 3D head shape, followed by a differential FLAME Layer as in DAD-3DNet(Martyniuk et al., 2022). Note that both branches have the same number of decoders.

Similar to the transformer in DETR (Carion et al., 2020) requiring $N$ zero-initialized object queries as the inputs of the decoder, the proposed method sets a corresponding number of input queries for each prediction task. For the main 3DMM parameters regression branch, we set $N$ to 413 indicating the number of 3DMM parameters. For the auxiliary 2D facial landmark detection branch, the value of $N$ is set to 68 to achieve the predictions of 68 landmark coordinates. Besides, the memory contains rich information related to the location of landmarks, while the landmark queries are data-dependent because their values need to be extracted from the memory during inference. Different from DETR where the queries are zero-initialized, we use a global average pooling layer and a fully connected layer to initialize the landmark queries with more meaningful values. Suppose the memory is denoted as $M$ with size $d \times h \times w$ and each query embedding is of size $d$, the initialization of queries can be computed as follows:

$$Q_{init} = FC(GlobalAvgPool(M)), \tag{1}$$

where $Q_{init}$ is of size $N \times d$.

The multi-task model is end-to-end trainable and the major goal is to predict the 3DMM parameters to reconstruct 3D head shape, while landmark detection is only an auxiliary task. Different from DAD-3DNet, which requires upsampling process and heatmap prediction using high-resolution feature maps, our model can achieve superior accuracy by using low-resolution feature maps. Besides, the structure of Transformers provides the information communication among the inputs, enhancing the robustness of the predictions through the relation among the more discriminative queries.

### 3.2 QUERY-AWARE MEMORY

Since using low-resolution feature maps and removing the encoder module, the framework is efficient but the accuracy is degraded. To address this trade-off problem, we propose a novel and lightweight module named QAMem based on an observation that the discriminative ability of the queries is limited within a specific grid of the memory.

Specifically, suppose there are three queries $Q_1$, $Q_2$, and $Q_3$, and the embedding value of the grid is denoted as $V_g$. In Figure 2(a), the resolution of the memory is $8 \times 8$ and the attention weights of

the three queries are 0.9, 0.1, and 0.3 respectively. Then the extracted values from the grid for each query are $0.9V_g$, $0.1V_g$, and $0.3V_g$ respectively, which means that the only difference of the three values is the scale. But the queries will act differently if the memory is of higher resolutions. As shown in Figure 2(b), the previous grid becomes a $4 \times 4$ sub-map when the memory is $32 \times 32$. Due to the increased resolutions, each query now is able to extract a different value from the grid. Can we also obtain such discriminative abilities on low-resolution feature maps? Yes, if the memory act differently to different queries. To achieve this, we compute $N$ new memories for $N$ queries from the original memory through $N$ corresponding convolutional layers. Figure 2(c) shows the mechanism that each query has a separate memory, which enables different queries to extract different values from the same grid on low-resolution feature maps. Note that the separate memories are only for value extractions, while the key of the queries is still shared.

Nevertheless, simply implementing the above method can be computationally heavy since there are usually tens of landmarks that require generating as many new memories. To address this, we propose an equivalent implementation that is much more efficient. Specifically, suppose $A$ is the attention weights with size $N \times S$, $M$ is the memory with size $S \times d$, $T_i$ with size $d \times d$ is the corresponding convolutional layer of $Q_i$, where $S$ denotes the number of feature map grids (i.e., $hw$). Then the extracted query $Q_i$ from the above method can be computed as follows:

$$Q_i = A^i_{1 \times S} \cdot M^i_{S \times d} = A^i_{1 \times S} \cdot (M_{S \times d} \times T^i_{d \times d}). \tag{2}$$

With our implementation, query $Q_i$ can be computed as:

$$Q_i = A^i_{1 \times S} \cdot (M_{S \times d} \cdot T^i_{d \times d}) = (A^i_{1 \times S} \cdot M_{S \times d}) \cdot T^i_{d \times d}. \tag{3}$$

Then the transform is only computed over the extracted query embedding rather than memory, which significantly reduces redundant computations and memories. In practice, the QAMem layer can be simply implemented as a convolutional layer with $1 \times 1$ kernel, 1 stride, and $N$ groups.

Additionally, QAMem module is applied to the facial landmark detection branch aiming to refine the location of landmarks through the position information in the feature map. Since the 3DMM parameters are not directly related to the spatial position in the feature map, QAMem module is not applicable in the 3DMM parameter regression branch.

### 3.3 MULTI-LAYER ADDITIVE RESIDUAL REGRESSION

Intuitively, predicting the residual coordinates based on the average coordinates is more reliable than randomly regressing the landmark coordinates. To improve the robustness of the predicted landmarks, the MARR is introduced in the decoder to guide the detection based on an average face model. As illustrated in Figure 3, the prediction of landmarks starts with the average model, and then each decoder predicts the residual coordinates from shallow to deep layer. Therefore, the MARR can aggregate multi-layer residual coordinates into a reliable initial reference to make the prediction easier. Specifically, the average face model $F_a$ is the initial reference which is obtained on the train set as follows:

$$F_a = \frac{1}{m} \sum_{i=1}^{m} lmk_i, \tag{4}$$

where, $m$ indicates the number of training samples and $lmk_i$ represents the ground-truth landmarks of $i_{th}$ image extracted from the ground-truth 3D vertices. Then, the predicted residual coordinates of each decoder will be added to the average face model and the final landmark coordinates $P_L$ can be computed as:

$$P_L = F_a + \sum_{i=1}^{n} res_i, \tag{5}$$

where, $n$ represents the number of decoder and $res_i$ indicates predicted residual coordinates of $i_{th}$ decoder.

### 3.4 MULTI-INFORMATION LOSS FUNCTION

We use the model excepting the QAMem and MARR modules as our baseline, where the loss function of baseline consists of three parts, including Landmark Regression Loss ($L_{lmk}$), 3D Head

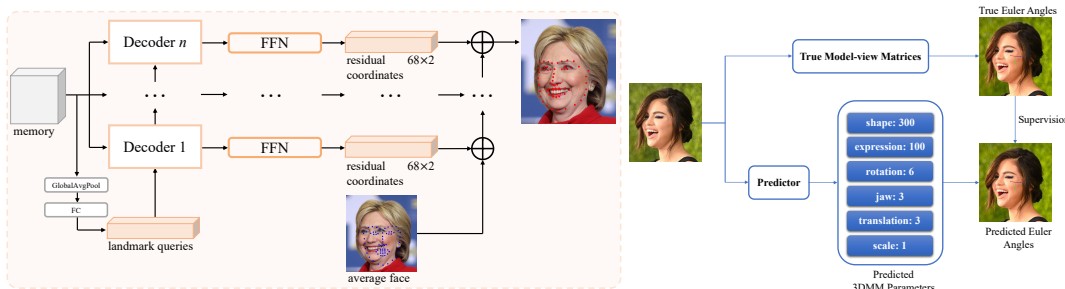

Figure 3: The structure of the MARR module.          Figure 4: The supervision of Euler angles.

Shape Loss ($L_{3D}$), and Reprojection Loss ($L_{reproj}$). The Landmark Regression Loss calculates the $Smooth_{L1}$ loss between the predicted and true 2D landmarks. For 3D Head Shape Loss, the 3D vertices can be computed by passing the predicted 3DMM parameters to a differentiable FLAME layer and only the set of "head" vertices is used in our task. Then 3D Head Shape Loss measures $L_2$ loss between normalized subsampled vertices of ground truth and predictions. For Reprojection Loss, the 2D vertices are obtained by reprojecting the 3D vertices onto the image and then subsampling the set of "head" vertices. We use the $Smooth_{L1}$ loss to measure the discrepancy between the reprojected subsampled vertices of ground truth and predictions.

In summary, the loss function of the baseline can be formulated as:

$$L_{base} = \lambda_1 L_{lmk} + \lambda_2 L_{3D} + \lambda_3 L_{rep}, \tag{6}$$

where $\lambda_1$, $\lambda_2$, and $\lambda_3$ are hyper parameters to balance each terms.

We observe that the predictive ability of the model is limited in the case of atypical head poses. To solve this issue, we introduce the Euler Angles Loss ($L_{euler}$) to train the model by providing more information supervision for the network.

### 3.4.1 Euler Angles Loss

As illustrated in Figure 4, the predicted Euler angles can be obtained from the predicted 3DMM parameters. To make full use of the annotations of the DAD-3DHeads dataset, the true Euler angles can be calculated by model-view matrices in the ground truth. Then, the Euler Angles Loss is introduced to calculate the $Smooth_{L1}$ loss between the predicted and true Euler angles as follows:

$$L_{euler} = smooth_{L1}(E_p - E_t), \tag{7}$$

where $E_p$ and $E_t$ indicate predicted and true Euler angles, respectively.

Finally, the overall optimal object for the proposed model can be formulated as follows:

$$L = \lambda_1 L_{lmk} + \lambda_2 L_{3D} + \lambda_3 L_{reproj} + \lambda_4 L_{euler}, \tag{8}$$

where $\lambda_4$ is hyper parameter, and we set $\lambda_1$, $\lambda_2$, $\lambda_3$, and $\lambda_4$ to 300.0, 50.0, 0.05, and 0.05.

## 4 Experiments

### 4.1 Experimental settings

#### 4.1.1 Implementation details

The proposed method is implemented in PyTorch and all of the experiments are conducted on a server with 1 NVIDIA V100 GPU. We use ResNet-50 (He et al., 2016) as the backbone which is initialized using the weights pre-trained on ImageNet (Deng et al., 2009). Adam (Kingma & Ba, 2015) is used as the optimizer and our model is trained for 360 epochs in total with a batch size of 32, where the differentiable FLAME layer is kept fixed. The learning rate is initialized to 0.0001 and then decayed by 10 at $240_{th}$ epoch, while the learning rate of the backbone is multiplied by 0.1. The hidden dimension $d$ in the decoder is set to 256.

Table 1: 3D dense head alignment on DAD-3DHeads benchmark.

| Method | Publication | NME ↓ | Z5 Acc. ↑ | Cham. Dis.↓ | Pose Err.↓ | FPS |
|---|---|---|---|---|---|---|
| RingNet | CVPR 2019 | 8.757 | 0.880 | 5.166 | 0.438 | - |
| 3DDFA-V2 | ECCV 2020 | 3.580 | - | 6.170 | 0.527 | - |
| DAD-3DNet | CVPR 2022 | 2.431 | 0.949 | **3.183** | 0.168 | 112 |
| **Trans3DHead** | - | **2.248** | **0.950** | 3.257 | **0.153** | 115 |

### 4.1.2 DATASETS

We use two public datasets to conduct the experiments, namely DAD-3DHeads and AFLW2000-3D dataset (Zhu et al., 2016). **DAD-3DHeads** is the state-of-the-art dataset for 3D dense head alignment, which contains rich annotations including extreme poses, facial expressions, challenging illuminations, and severe occlusion cases. It is an in-the-wild facial landmark dataset containing 37,840 training images, 4,312 validation images, and 2,746 test images. **AFLW2000-3D dataset** includes the first 2,000 samples from the in-the-wild AFLW dataset (Köstinger et al., 2011) and each of these samples is labeled with the ground truth of 3D face and the corresponding 68 landmarks.

### 4.1.3 EVALUATION METRIC

On DAD-3DHeads dataset, the NME, Z5 Accuracy, Chamfer Distance, and Pose Error proposed in (Martyniuk et al., 2022) are calculated to measure the goodness-of-fit for the 3D dense head alignment task. Specifically, the NME is computed on 68 landmarks which measures the normalized mean error of the predictions. Z5 Accuracy evaluates the ordinal distance of the Z-coordinate and is calculated only on the vertices of the "head" subset. Chamfer distance is a one-sided metric calculated from ground truth mesh to the predicted one, in which the vertices are first aligned by seven key points correspondences (Sanyal et al., 2019), and then only vertices of the "face" subset are used to compute the distances. Pose Error measures the accuracy of pose predictions. On AFLW2000-3D dataset, we calculate the mean absolute error (MAE) of predicted Euler angles.

## 4.2 QUANTITATIVE EVALUATION

To fully illustrate the effectiveness of the proposed method, we present the quantitative comparison with the advanced methods on 3D dense head alignment task and 3D head pose estimation task.

### 4.2.1 3D DENSE HEAD ALIGNMENT

The proposed model is compared with 3 advanced 3DMM-based methods (Guo et al., 2020; Sanyal et al., 2019; Martyniuk et al., 2022), and the quantitative results on the full test dataset of DAD-3DHeads (Martyniuk et al., 2022) are shown in Table 1. The DAD-3DNet is evaulated with author's code and released model, and our method is also based on the same code base for training and evaluation. The proposed method achieves the best scores in terms of NME, Z5 Accuracy, and Pose Error, outperforming 3DDFA-V2 and RingNet on all four metrics. This improvement indicates that our model is more robust in facial landmark detection.

### 4.2.2 3D HEAD POSE ESTIMATION

As presented in Table 2, the proposed model is compared with 10 advanced methods (Bulat & Tzimiropoulos, 2017a; King, 2009; Doosti et al., 2020; Deng et al., 2020; Albiero et al., 2021) including 5 3DMM-based methods (Zhu et al., 2019; Guo et al., 2020; Sanyal et al., 2019; Wu et al., 2021; Martyniuk et al., 2022). In overall MAE, our model achieves leading performance among all models except SynergyNet. However, our model achieves the best MAE in the estimation of Yaw, outperforming all other state-of-the-art methods by a significant margin. It is also worth noting that our model surpasses DAD-3DNet on all four metrics, which indicates the superiority of our algorithm. Besides, our model obtains more balanced performance across all angles compared to other methods, demonstrating the effectiveness of the multi-information supervision with Euler Angles Loss.

Table 2: 3D head pose estimation results on AFLW2000-3D dataset.

| Method | Publication | MAE↓ | Pitch↓ | Roll↓ | Yaw↓ |
|--------|-------------|------|--------|-------|------|
| Dlib | JMLR 2009 | 13.29 | 12.60 | 9.00 | 18.27 |
| FAN | ICCV 2017 | 9.12 | 12.28 | 8.71 | 6.36 |
| 3DDFA | TPAMI 2017 | 7.39 | 8.53 | 7.39 | 5.40 |
| RingNet | CVPR 2019 | 8.27 | 4.39 | 13.51 | 6.92 |
| HopeNet | CVPR 2020 | 6.16 | 6.56 | 5.44 | 6.47 |
| RetinaFace | CVPR 2020 | 6.22 | 9.64 | 3.92 | 5.10 |
| 3DDFA-V2 | ECCV 2020 | 7.56 | 8.48 | 9.89 | 4.30 |
| Img2Pose | CVPR 2021 | 3.91 | 5.03 | 3.28 | 3.43 |
| SynergyNet | 3DV 2021 | **3.35** | **4.09** | **2.55** | 3.42 |
| DAD-3DNet | CVPR 2022 | 3.66 | 4.76 | 3.15 | 3.08 |
| **Trans3DHead** | - | 3.38 | 4.39 | 2.84 | **2.91** |

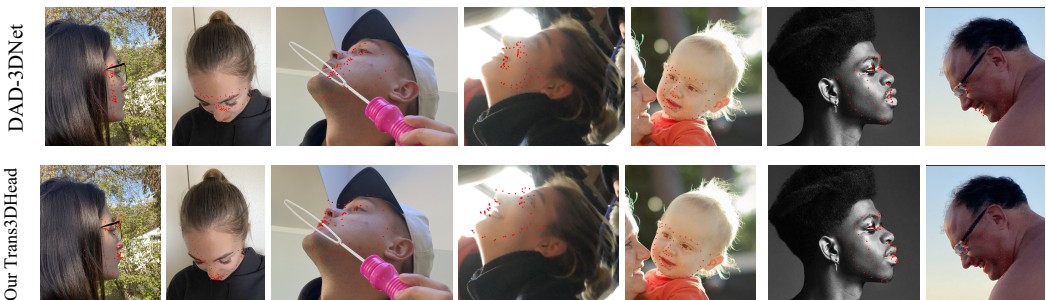

Figure 5: Qualitative comparison on challenging cases from DAD-3DHeads benchmark.

## 4.3 QUALITATIVE EVALUATION

To visually display the facial detection results, we select some challenging cases from the DAD-3DHeads test set for comparison. Figure 5 shows the visualizations of 68 landmarks predicted by DAD-3DNet and the proposed Trans3DHead. It can be seen that our Trans3DHead performs better in the mouth, face contour regions, and atypical head poses. It further indicates that the multi-information supervision with Euler Angles Loss is effective to atypical poses and the information exchange among different vertices is helpful to accurate localization.

Additionally, some failure cases are presented in Figure 6. It can be seen that both DAD-3DNet and our Trans3DHead are limited to severe occlusions, near horizontal or vertical flips, and the back side of the head with unseen facial features. However, the estimations of the head poses by our Trans3DHead are more reliable than DAD-3DNet.

## 4.4 ABLATION STUDY

Taking the evaluation under the validation set of DAD-3DHeads as an example, we conduct the ablation experiments. For a fair comparison, the parameters for all experiments are set to the same in both training and testing.

### 4.4.1 COMPONENT EFFECTIVENESS

As presented in Table 3, each component in the proposed method is added one by one to evaluate the efficacy, where the models without MARR directly predict landmarks by last decoder. With the help of QAMem, the third model achieves better NME than the baseline. It demonstrates that utilizing separate memories is effective to improve accuracy. By adding the Euler Angles Loss to the third model, the fourth model is superior to the third model in terms of Z5 Accuracy and Pose Error. These results verify that the multi-information supervision with Euler Angles Loss is beneficial to accurately estimate the head pose. Furthermore, we integrate three components together to obtain

Table 3: Left: Component effectiveness on validation set. QA indicates QAMem, and Euler indicates Euler Angles Loss. Right: The analysis of encoder and decoder layers on validation set where the structure of encoder is adapted from DETR. E and D indicate encoder and decoder respectively.

| Method | NME ↓ | Z5 Acc. ↑ | Cham. Dis.↓ | Pose Err.↓ | E/D | NME ↓ | Z5 Acc. ↑ | Cham. Dis.↓ | Pose Err.↓ |
|---|---|---|---|---|---|---|---|---|---|
| DAD-3DNet | 1.956 | 0.9571 | **2.749** | 0.130 | 0/1 | 1.869 | 0.9578 | 2.819 | 0.128 |
| baseline | 1.847 | 0.9578 | 2.791 | 0.126 | 0/2 | 1.831 | 0.9578 | **2.803** | 0.128 |
| +QA | 1.831 | 0.9578 | 2.803 | 0.128 | 0/3 | **1.821** | 0.9580 | 2.816 | 0.126 |
| +Euler+QA | 1.837 | 0.9580 | 2.819 | 0.127 | 1/2 | 1.836 | 0.9582 | 2.827 | 0.125 |
| +MARR+QA | 1.851 | 0.9572 | 2.802 | 0.129 | 2/2 | 1.835 | 0.9574 | 2.840 | 0.127 |
| Trans3DHead | **1.822** | **0.9585** | 2.796 | **0.123** | 3/2 | 1.828 | **0.9585** | 2.849 | **0.124** |

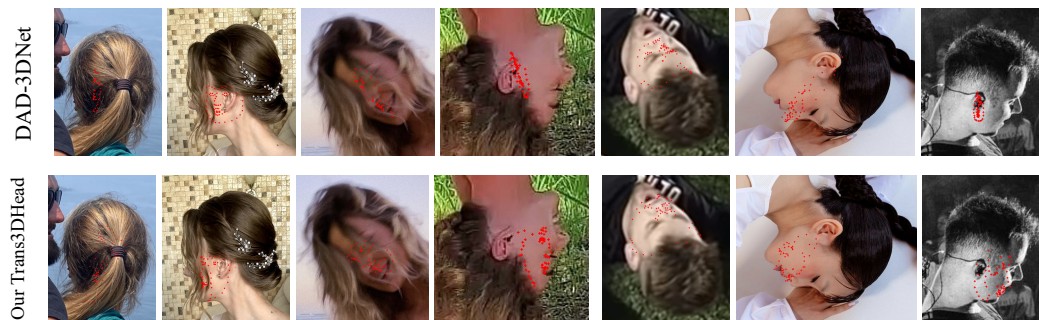

Figure 6: Failure cases from DAD-3DHeads benchmark.

our Trans3DHead which achieves the best results on NME, Z5 Accuracy, and Pose Error. Since the average face may have some limitation in handling large head pose variations, the effectiveness of MARR is more observed when the Euler Angle Loss is applied to improve the performance under large pose variations, that is, from the fourth model to Trans3DHead.

### 4.4.2 ENCODER AND DECODER LAYERS

To observe the effectiveness of the encoder and decoder, the baseline model with QAMem is evaluated under the different numbers of encoder and decoder layers. As presented in Table 3, stacking more decoder layers improves the performance while more encoder layers have an inconspicuous effect or even reduce the results. We speculate that facial landmark detection primarily focuses on the localization of single points, so the high-level semantic feature extraction of encoder layers may introduce spatial noises, hurting the localization of points. However, more decoder layers bring more computational costs. To balance the accuracy and efficiency, we remove the encoder layer and use two decoder layers for our Trans3DHead.

## 5 CONCLUSION

In this work, we propose an efficient multi-task 3D head alignment network named Trans3DHead. With two task-oriented regression branches based on transformers, the model enhances the information communication among queries and is suitable for various 3D head alignment tasks. The proposed QAMem removes the dependence on high-resolution feature maps, which is efficient and effective to make use of low-resolution feature maps by utilizing separate memories. The MARR module can achieve more stable and reliable location of the facial landmarks by adding the multi-layer residual coordinates to an average reference. Besides, adding the Euler Angles Loss to the original multi-information loss function enhances the robustness of the model to atypical head poses. Our method achieves better performance on public benchmarks in terms of key metrics. In the future, we hope to explore more effective manners based on different head poses to dynamically provide the references of the average model.

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

## A  APPENDIX

The outline of the Appendix is as follows:

- Further ablation study.
- More visual examples on DAD-3DHeads benchmark.
- More visual examples on AFLW2000-3D dataset.

### A.1  FURTHER ABLATION STUDY

Taking the evaluation under the validation set of DAD-3DHeads dataset as an example, we conduct further ablation experiments to analyze the fixed average model, the effect of multi-layer additive residual regression (MARR), the query-aware memory (QAMem), the memory-related query initialization (MQinit) used in the facial landmark detection branch, the effectiveness of the multi-task framework, and the effect of different 3DMM parameter settings.

### A.1.1  FIXED OR LEARNABLE AVERAGE MODEL

As shown in Table 4, we conduct experiments with the average model fixed and learnable respectively for the refined facial landmark detection. For the two learnable average models, the values of average face are treated as the learnable parameters, where one model is initiated with the average face calculated by Eq.(4), and the other is randomly initiated. It can be seen that the fixed average

Table 4: Average model analysis on validation set.

| Method | NME↓ | Z5 Acc. ↑ | Cham. Dis.↓ | Pose Err.↓ |
|---|---|---|---|---|
| Trans3DHead | **1.822** | **0.9585** | **2.796** | **0.123** |
| +averinit | 1.849 | 0.9584 | 2.816 | 0.125 |
| +randinit | 1.842 | 0.9582 | 2.797 | 0.126 |

Table 5: Further analysis of QAMem on validation set of DAD-3DHeads dataset.

| Method | NME↓ | Z5 Acc. ↑ | Cham. Dis.↓ | Pose Err.↓ |
|---|---|---|---|---|
| baseline | 1.847 | 0.9578 | 2.791 | 0.126 |
| +QAMem-1 | **1.831** | 0.9578 | 2.803 | 0.128 |
| +QAMem-2 | **1.831** | **0.9581** | **2.782** | **0.125** |

model surpasses two learnable average models on all four metrics. Our analysis suggests that the learnable average models may introduce more parameters and more uncertainties which makes the regression of landmark coordinates more challenging, while the fixed average model provides more stable and reliable references.

### A.1.2 MULTI-LAYER ADDITIVE RESIDUAL REGRESSION

As illustrated in Figure 7, after aggregating multi-layer residual coordinates into a reliable initial reference, the landmarks visualization becomes more and more accurate from shallow to deep layer. Besides, we normalize residual coordinates by the head bounding box size to reduce the effect of the face scale. Then the distribution of absolute values of normalized residual coordinates for each decoder layer is visualized as in Figure 8. The visual results further indicate that as the decoder layer deepens, more residual coordinates gradually approach 0, becoming sparse, which realizes refining the landmarks layer-by-layer and makes the prediction easier.

### A.1.3 QUERY-AWARE MEMORY

As presented in Table 5, the baseline indicates the model excepting the QAMem, MARR, and Euler Angles Loss. Based on the baseline, the second model uses QAMem only in the facial landmark detection branch, while the third model applies QAMem to both the facial landmark detection branch and the 3DMM parameters regression branch. It can be seen that the second model surpasses the baseline in terms of NME, while the third model shows no significant improvement in NME. Besides, the third model is more memory-consuming than the second one, because the QAMem module in 3DMM regression requires about six times as many parameters as it does in landmark regression. Therefore, the QAMem module is only used in the facial landmark detection branch in our Trans3DHead.

### A.1.4 MEMORY-RELATED QUERY INITIALIZATION

The MQinit indicates the global average pooling layer and the fully connected layer in Figure 1 of our paper used on memory to initialize the landmark queries in the facial landmark detection branch. As shown in Table 6, the first model represents the Trans3DHead without MQinit, the second model is our Trans3DHead using MQinit only in the facial landmark detection branch, and the third model uses MQinit both in the facial landmark detection branch and the 3DMM parameters regression branch. With the help of MQinit, the second model is effective to boost the accuracy of facial landmark detection, while using the MQinit in the 3DMM parameters regression branch shows no significant improvement in NME. Therefore, the MQinit module is only used in the facial landmark detection branch to make full use of the rich location information contained in the memory.

### A.1.5 MULTI-TASK FRAMEWORK

In order to explore the effect of the multi-task framework, we conduct a comparative experiment between the Trans3DHead with only 3DMM parameters regression branch and the Trans3DHead.

Table 6: Further analysis of MQinit on validation set of DAD-3DHeads dataset.

| Method | NME ↓ | Z5 Acc. ↑ | Cham. Dis.↓ | Pose Err.↓ |
|---|---|---|---|---|
| noMQinit | 1.865 | 0.9577 | 2.795 | 0.127 |
| +MQinit-1 | 1.822 | **0.9585** | 2.796 | **0.123** |
| +MQinit-2 | **1.821** | 0.9577 | **2.781** | 0.125 |

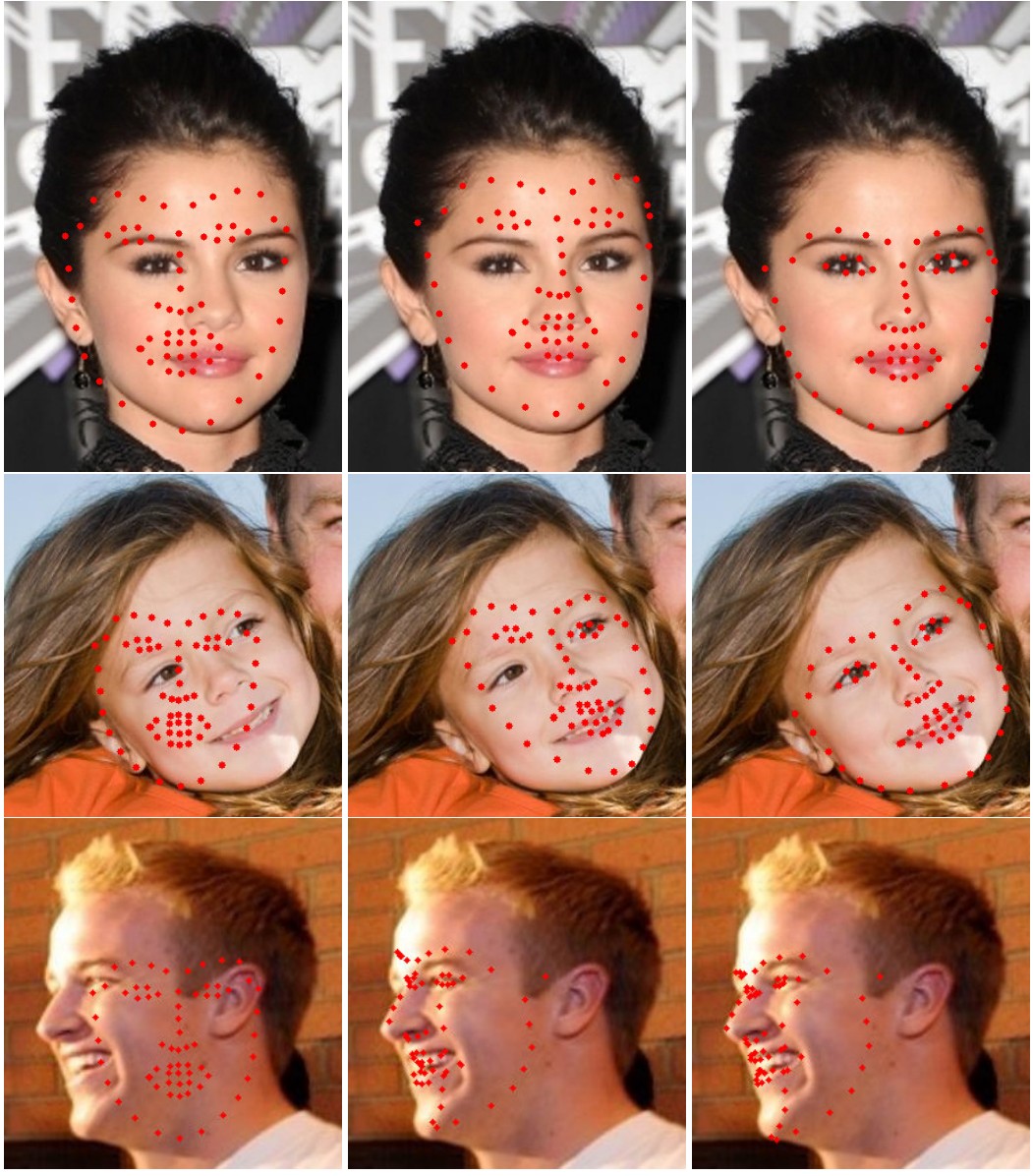

Figure 7: The landmarks visualization from the MARR module on validation set of DAD-3DHeads dataset. Left to right: initial average face, the outputs from the first decoder layer, the outputs from the second decoder layer.

As presented in Table 7, the multi-task Trans3DHead effectively improves the performance of the single-task model on 3D head alignment.

Table 7: Further analysis of multi-task framework on validation set of DAD-3DHeads dataset. Note that the only-3DMM indicates the Trans3DHead with only 3DMM parameters regression branch.

| Method | NME ↓ | Z5 Acc. ↑ | Chamfer Dis.↓ | Pose Err.↓ |
|--------|-------|-----------|---------------|-----------|
| only-3DMM | - | 0.9567 | 2.839 | 0.133 |
| Trans3DHead | 1.822 | **0.9585** | **2.796** | **0.123** |

Table 8: The ablation study of different 3DMM parameter settings on validation set of DAD-3DHeads dataset.

| | DAD-3DNet | | | | Trans3DHead | | | |
|---|---|---|---|---|---|---|---|---|
| | NME ↓ | Z5 Acc. ↑ | Chamfer Dis.↓ | Pose Err.↓ | NME ↓ | Z5 Acc. ↑ | Chamfer Dis.↓ | Pose Err.↓ |
| number = 60 | 2.354 | 0.9481 | 3.298 | 0.153 | 1.886 | 0.9529 | 3.241 | 0.129 |
| number = 160 | 2.312 | 0.9518 | 2.946 | 0.145 | 1.901 | 0.9560 | 2.845 | 0.130 |
| number = 413 | **1.956** | **0.9571** | **2.749** | **0.130** | **1.822** | **0.9585** | **2.796** | **0.123** |

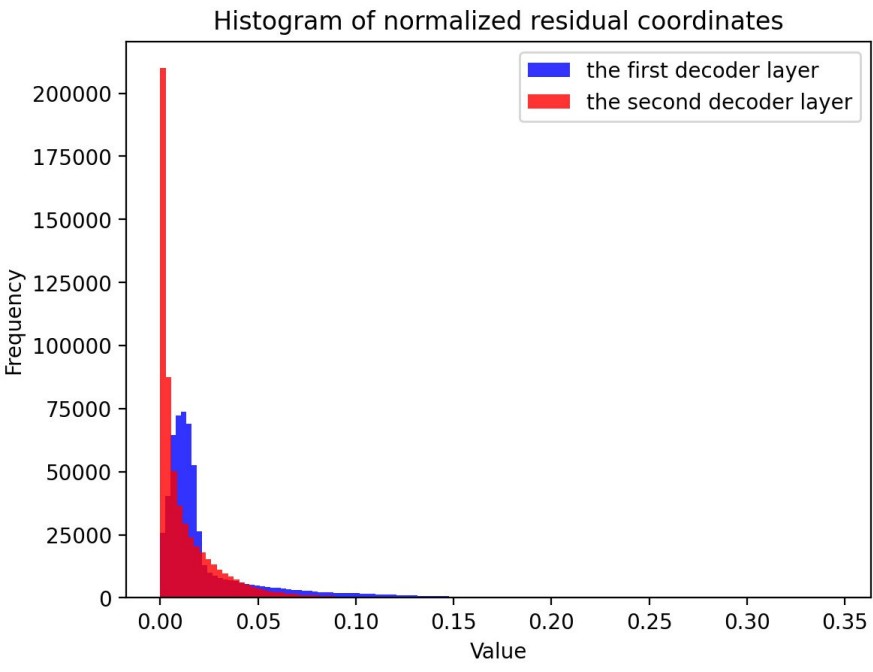

Figure 8: Histogram of normalized residual coordinates from each decoder layer on validation set of DAD-3DHeads dataset.

### A.1.6  DIFFERENT 3DMM PARAMETER SETTINGS

The 3DMM parameters used in DAD-3DNet and the proposed Trans3DHead are adapted to FLAME, which are extensions of the standard 3DMM parameters. The number of the extended 3DMM parameters used in the final Trans3DHead is 413, including 300 shape parameters, 100 expression parameters, 3 jaw parameters, 6 rotation parameters, 3 translation parameters, and 1 scale parameter to control FLAME. We also conduct experiments with two additional 3DMM parameter settings. The first one is with the same number of 3DMM parameters as in SynergyNet (40 shape parameters, 10 expression parameters without jaw parameter), while the second one is used in RingNet (100 shape parameters, 50 expression parameters without jaw parameter). Keeping the original setting of 6 rotation parameters, 3 translation parameters, and 1 scale parameter in FLAME,

we use two parameter settings of 60 parameters and 160 parameters to train our Trans3DHead and DAD-3DNet on the DAD-3DHeads dataset, respectively. As shown in Table 8, regressing more parameters is helpful for better reconstructing 3D head shape for both DAD-3DNet and the proposed Trans3DHead. Besides, the proposed Trans3DHead is superior to DAD-3DNet in all three parameter settings. When only regressing 60 parameters, our model shows more significant advantages than DAD-3DNet, which also indicates that the cross-attention mechanism is effective to achieve the information communication among different elements.

## A.2 MORE VISUAL EXAMPLES ON DAD-3DHEADS BENCHMARK

To further demonstrate the superior performance of our model, we further display the dense landmarks visualization and mesh visualization of DAD-3DNet and our model on DAD-3DHeads benchmark for comparison.

### A.2.1 DENSE LANDMARKS VISUALIZATION

As illustrated in Figure 9, both DAD-3DNet and the proposed Trans3DHead allow for flexibly choosing the desired landmark subset after the entire vertices are predicted from 3DMM parameters. Our model performs better than DAD-3DNet on 68 landmarks prediction, thanks to the multi-layer additive residual regression module starting from average facial landmarks. Moreover, our model also recovers more accurate 3D head geometry compared with DAD-3DNet.

### A.2.2 MESH VISUALIZATION

As shown in Figure 10, the face mesh and 3D head mesh can be generated from the 3DMM parameters predicted by DAD-3DNet and the proposed Trans3DHead. Compared with DAD-3DNet, our model can handle the atypical head poses better and generates more accurate meshes. These visualization results further verify the effectiveness of the proposed model. Besides, It can be clearly observed that our method not only obtains more accurate shapes than SynergyNet but also reconstructs the shape of entire face, the full 3D head, and neck, while SynergyNet can only reconstruct face shape, which further illustrates the difference between the SynergyNet and our Trans3DHead.

## A.3 MORE VISUAL EXAMPLES ON AFLW2000-3D DATASET

### A.3.1 HEAD POSE VISUALIZATION

The proposed Trans3DHead achieves comparable results with SynergyNet on AFLW2000-3D dataset in the head pose estimation task. Figure 11 shows some examples of head pose estimation predicted by the SynergyNet and the proposed Trans3DHead on the AFLW2000-3D dataset. It can be observed that the SynergyNet performs slightly better in predicting pitch and roll, while the proposed model has more advantages in handling side faces.

### A.3.2 LANDMARKS VISUALIZATION

We further conduct qualitative comparison on AFLW2000-3D dataset to visually display the facial landmark detection results predicted by SynergyNet, DAD-3DNet, and the proposed Trans3DHead. As shown in Figure 12, the proposed Trans3DHead performs better in the mouth, face contour regions, and atypical head poses.

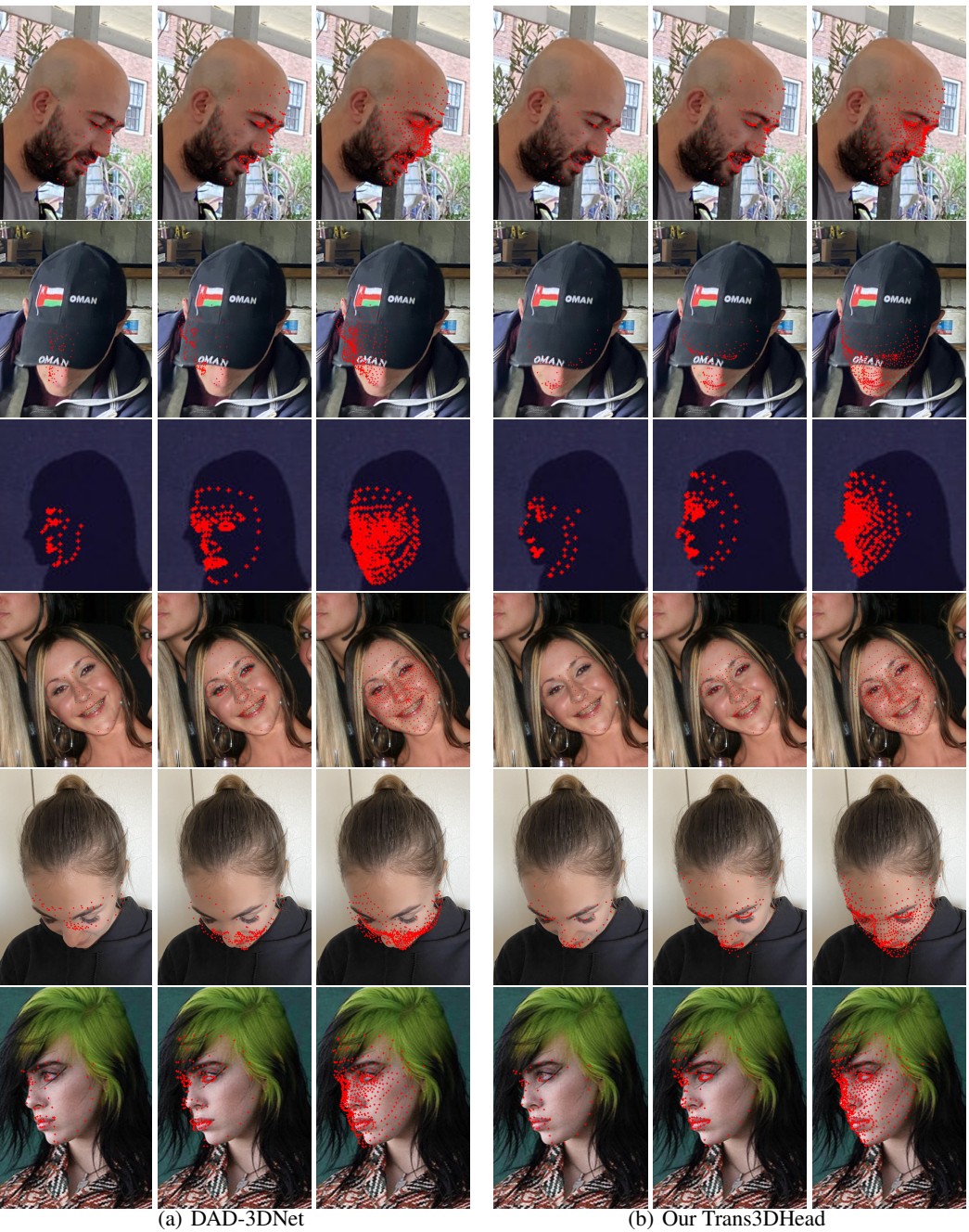

(a) DAD-3DNet            (b) Our Trans3DHead

Figure 9: Dense landmarks visualization from DAD-3DHeads benchmark. Left to right in each subfigure: 68 landmarks, 191 landmarks, 445 landmarks. Note that both DAD-3DNet and our model predict the 68 landmarks from the 2D branch directly, while 191 landmarks and 445 landmarks are generated by reprojecting the 3D vertices.

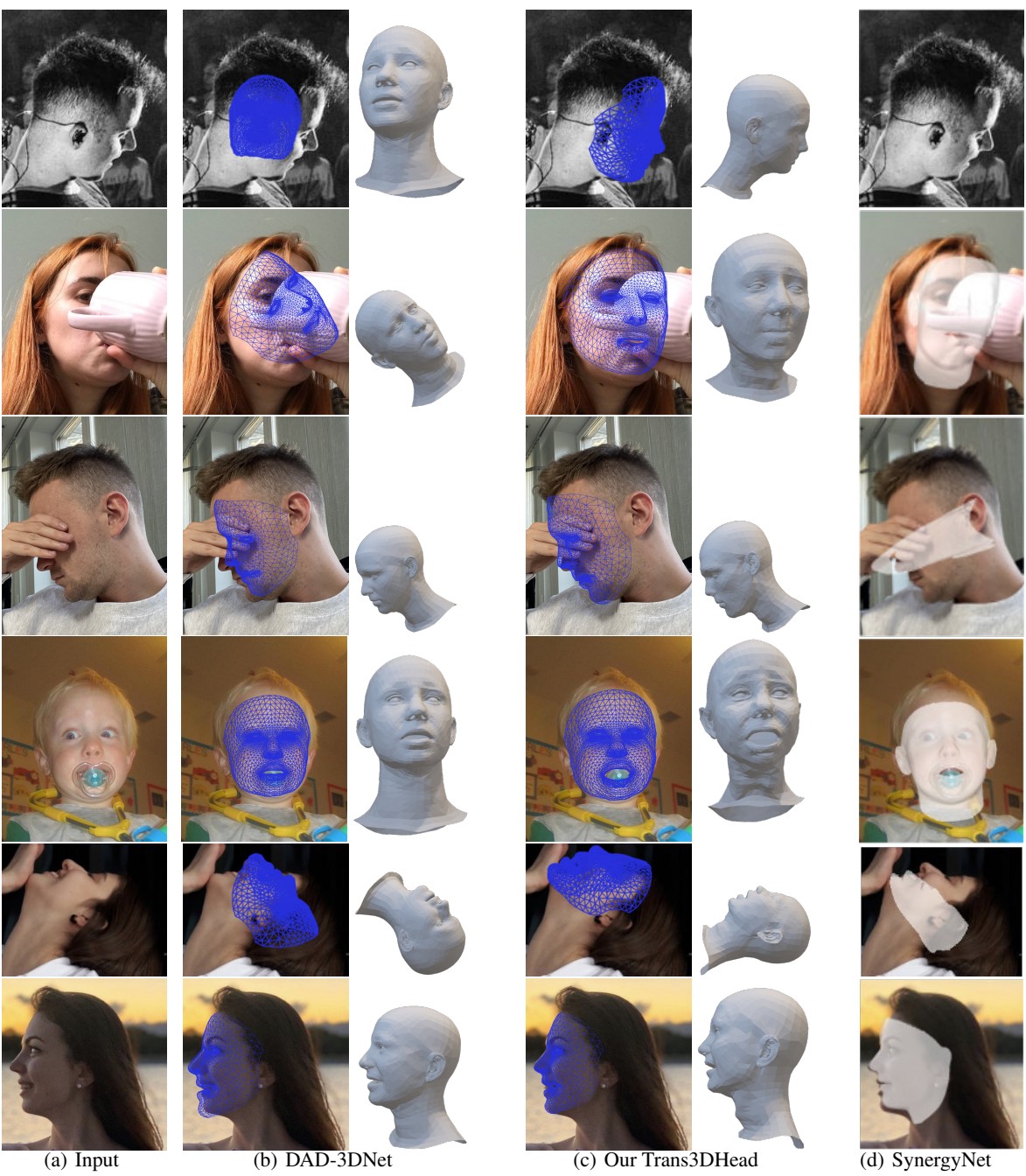

(a) Input                    (b) DAD-3DNet                    (c) Our Trans3DHead                    (d) SynergyNet

Figure 10: Mesh visualization from DAD-3DHeads benchmark. (a): input image; Left to right in (b) and (c): face mesh, 3D head mesh; (d): face mesh.

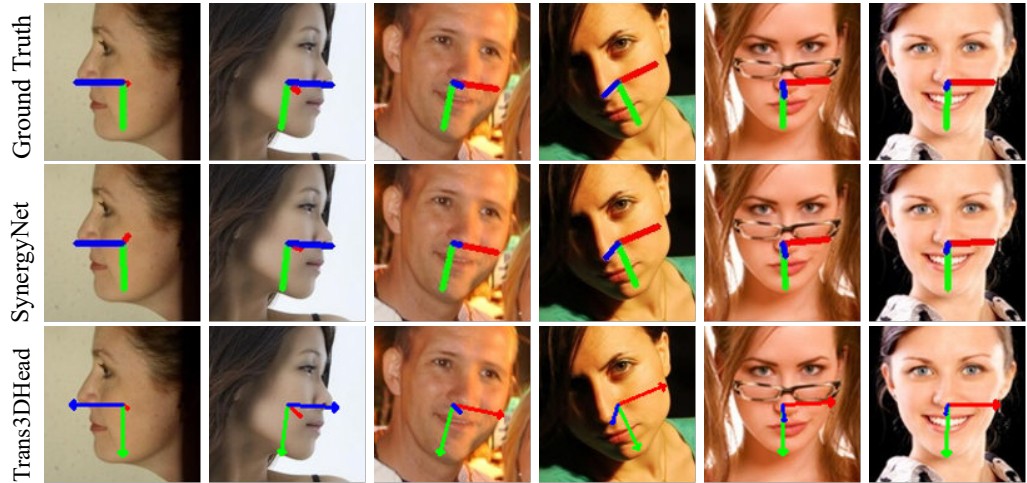

Figure 11: Examples of head pose estimation predicted by the SynergyNet and the proposed Trans3DHead on the AFLW2000-3D dataset.

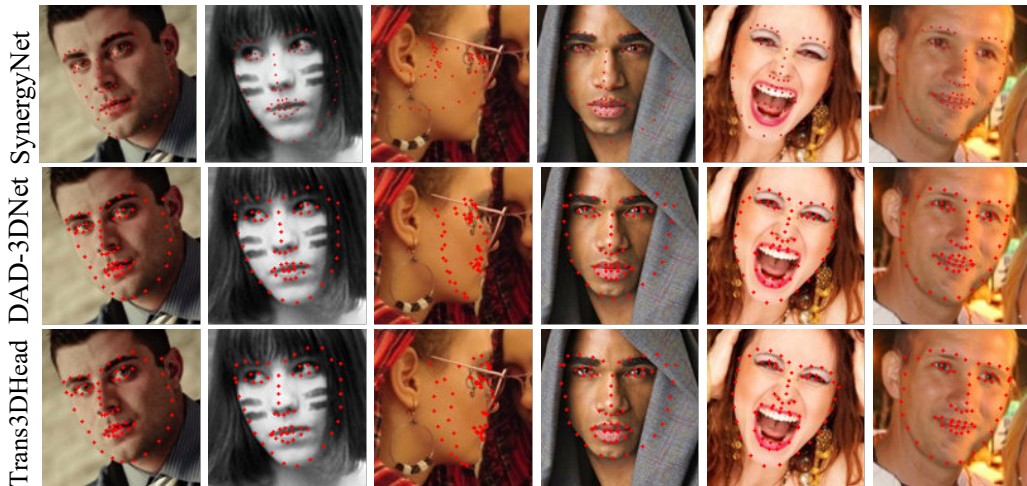

Figure 12: Examples of facial landmarks predicted by the SynergyNet, the DAD-3DNet, and the proposed Trans3DHead on the AFLW2000-3D dataset.

