# OpenReview forum: "An Efficient Multi-Task Transformer for 3D Face Alignment"
_ICLR.cc/2024/Conference — Submitted to ICLR 2024_

### Official Review · Reviewer_PGZe · 2023-10-31

**Soundness:** 2 fair
**Presentation:** 1 poor
**Contribution:** 1 poor
**Rating:** 5
**Confidence:** 4

**Summary:**

Authors proposed the multi-task framework based on the transformer architecture to efficiently capture the high-resolution information in facial landmark detection task. Query-aware memory module is newly introduced and the multi-layer additive residual regression module and Euler angles loss are proposed. Experiments on two public benchmarks show the effectiveness of the proposed method showing the SOTA results.

**Strengths:**

QAMem module looks sound and it is well presented in the Figure 2.
MARR and Euler angles loss also look effective to tackle the targetted problem.
English and presentation are sufficiently good to understand the work.

**Weaknesses:**

Less qualitative results: only 1 dataset is used for qualitative results. More is required.
Even though the authors insist that the proposed method is efficient; while there is no report for the time complexity in their results.
In ablation study, when comparing Euler+QA and MARR+QA, the accuracy improvement is quite limited. It is unclear the accuracy improvement was due to the components, or not.
Explanations for some equations are rather blurry (for eqs 2 and 3).

**Questions:**

There is less explanation for how to proceed the equations from 2 to 3.
Why in ablation study, the three combination only can improve the final accuracy, even though partial combination is not that effective?

---

> ### Author Response · Authors · 2023-11-22
> **Responses to Reviewer PGZe**
>
> ## [Question 1]
> Less qualitative results: only 1 dataset is used for qualitative results. More is required.
> ## [Answer]
> Thank you for your comment. We have added qualitative results on AFLW2000-3D dataset to visually display the facial landmark detection results predicted by SynergyNet, DAD-3DNet, and the proposed Trans3DHead. As shown in Figure 12 in the appendix, our Trans3DHead performs better in the mouth, face contour regions, and atypical head poses.
>
> ## [Question 2]
> Even though the authors insist that the proposed method is efficient; while there is no report for the time complexity in their results.
> ## [Answer]
> Thank you for the comment. Please refer to the Common Questions and Responses 2.
>
>
> ## [Question 3]
> In ablation study, when comparing Euler+QA and MARR+QA, the accuracy improvement is quite limited. It is unclear the accuracy improvement was due to the components, or not. Why in ablation study, the three combination only can improve the final accuracy, even though partial combination is not that effective?
> ## [Answer]
> Thank you for your comment. Since each component is proposed to address a specific problem, the accuracy improvement is limited when each component is used independently. However, when considering the comprehensive problems, the combination of all the components will achieve better results. Specifically, the Euler angle loss is only proposed for head pose estimation, so comparing Euler+QA to QA, it only improves the pose error and Z5 accuracy of the entire 3D head, but it does not improve the landmark detection and chamfer distance for face shape, as 3D head is quite related to head pose, but face shape is relatively not. As for MARR, as commented by Reviewer QjYA, the average face may have some limitation in handling large head pose variations, therefore, its effectiveness is more observed when the Euler angle loss is applied to improve the performance under large pose variations, that is, from Euler+QA to Trans3DHead (Euler+QA+MARR). We will make this point clearer in the paper.
>
> ## [Question 4]
> Explanations for some equations are rather blurry (for eqs 2 and 3). There is less explanation for how to proceed the equations from 2 to 3.
> ## [Answer]
> Thank you for pointing out this. We have updated the Eq. (2) and Eq. (3) to make it easier to understand. The Eq. (2) is the original calculation form of the QAMem module, which represents that each query corresponds to a separate memory rather than the shared memory. In order to reduce computations in practice, according to the associative law of multiplication, the operation order of multiplication can be changed to the form of Eq. (3), with the advantage that A $\times$ M can be precomputed only once, resulting in only one matrix multiplication for each query i, in contrast to two matrix multiplications for each query i in Eq. (2). This significantly reduces redundant computations and memories.
>
> Specifically, with Eq. (2) the complexity in terms of multiplications is O(NSd + NSdd)), while that of Eq. (3) is O(NSd + Ndd). Therefore, Eq. (3) reduces the computation of O(Ndd(S-1)), which is significant especially when N=68 and S=8 $\times$ 8 in the Trans3DHead model.

---

### Official Review · Reviewer_zzMV · 2023-10-31

**Soundness:** 2 fair
**Presentation:** 3 good
**Contribution:** 2 fair
**Rating:** 5
**Confidence:** 4

**Summary:**

This paper proposes a multi-task 3D face alignment framework based on a transformer. The objective is to overcome three main drawbacks of existing methods: (i) first, the 2D facial landmark detection task and the 3DMM parameters prediction task are parallelized in the form of two transformer branches. 3DMM parameters are regressed through Transformers, where the cross-attention mechanism is used to enhance the information communication among task-oriented queries and extracted feature maps in the designed decoder; (ii) A lightweight module named query-aware memory (QAMem) is proposed that makes up the accuracy loss from lower feature map resolutions. To enhance the robustness of the predicted landmarks, the average vertices coordinates of the training set are calculated, then a multi-layer additive residual regression (MARR) module is designed in the decoder to guide the detection under the reference of an average face model. (iii) A multi-information loss function is used to optimize the network.

**Strengths:**

The main contributions are:
- A Transformer-based multi-task framework is proposed for 3D face alignment, using a multi-task structure. 3DMM parameters are regressed through Transformers, where the cross-attention mechanism achieves the information communication among different elements.
- A Euler Angles Loss in introduced to the multi-information loss function for network optimization, which enhances the predictive ability in the case of atypical head poses.

**Weaknesses:**

- The title of the paper is not appropriate in my opinion. The title emphasizes a face alignment contribution, while the content focuses more on face landmarks detection.
- Table 2 indicates results that are comparable to the state-of-the-art but for some cases do not improve on it.
- Parameters of a 3DMM are regressed in this work. However, it is not clear how much the choice of the 3DMM impacts on the results. Did the authors try with different 3DMMs?
- In the ablation study it is not clear the impact of the 3DMM on the results.
- An analysis of the computational cost of the method in comparison with other solutions is missing. For landmarks detection the capacity of the approach to work in real time is important. Authors should clarify this point.

---------
Thanks to the authors for the answers and the revision of the paper.
I have read the other reviews and the authors’ rebuttal and also checked the new material included in the paper. Overall, I think the most substantial weaknesses of this work are still there. The novelty remains limited as well as the experimental evaluation is not fully convincing (quantitative results only for one dataset, performance comparable but in some case below the sota). Based on this, I think the contribution of this paper is below the ICLR standard, so I keep my original rating.

**Questions:**

Q1: The contribution of the used 3DMM on hte final results is not clear. Did the authors try with different 3DMMs? Can they show results using different 3DMMs?
Q2: In the ablation study it is not clear the impact of the 3DMM on the results.
Q3: Authors should discuss and illustrate the computational cost of the method in comparison with other solutions is missing.

---

> ### Author Response · Authors · 2023-11-22
> **Responses to Reviewer zzMV**
>
> ## [Question 1]
> The title of the paper is not appropriate in my opinion. The title emphasizes a face alignment contribution, while the content focuses more on face landmarks detection.
> ## [Answer]
> Our main task is 3D alignment, while landmark detection is only an auxiliary task. The major goal is to predict the 3DMM parameters to reconstruct 3D head shape. As shown in Figure 1, our model mainly consists of a 3DMM parameter regression branch and an auxiliary facial landmark detection branch. The 3D alignment relies on 3DMM parameters regression branch which can regress a set of 3DMM parameters and reconstruct the 3D head shape by a differentiable FLAME decoder. In the paper, Figure 1 shows the overall 3D alignment pipeline, Sec. 3.1 introduces the proposed 3D alignment framework, and Sec. 3.4 describes the overall losses for both landmark detection, as well as 3D shape reconstruction and reprojection. Besides, the Euler angle loss is proposed in Sec. 3.4.1 for improving the 3D alignment via 3D pose estimation. Thank you for your comment. We will further improve the presentation of Sec. 3.1 for the whole 3D alignment framework.
>
> ## [Question 2]
> Table 2 indicates results that are comparable to the state-of-the-art but for some cases do not improve on it.
> ## [Answer]
> Thank you for the comment. Please refer to the Common Questions and Responses 1 for the comparison to SynergyNet.
>
> ## [Question 3]
> Parameters of a 3DMM are regressed in this work. However, it is not clear how much the choice of the 3DMM impacts on the results. Did the authors try with different 3DMMs? In the ablation study it is not clear the impact of the 3DMM on the results. The contribution of the used 3DMM on hte final results is not clear. Did the authors try with different 3DMMs? Can they show results using different 3DMMs? In the ablation study it is not clear the impact of the 3DMM on the results.
> ## [Answer]
> Thank you for your valuable suggestion. The 3DMM parameters used in DAD-3DNet and the proposed Trans3DHead are adapted to FLAME, which are extensions of the standard 3DMM parameters. The number of the extended 3DMM parameters used in the final Trans3DHead is 413, including 300 shape parameters, 100 expression parameters, 3 jaw parameters, 6 rotation parameters, 3 translation parameters, and 1 scale parameter to control FLAME, which allows to use the first few parameters of the 300 shape parameters, 100 expression parameters, and 3 jaw parameters to express the shape and expression.
>
> We have conducted experiments with two additional 3DMM parameter settings. The first one is with the same number of 3DMM parameters as in SynergyNet (40 shape parameters, 10 expression parameters without jaw parameter), while the second one is used in RingNet (100 shape parameters, 50 expression parameters without jaw parameter). Keeping the original setting of 6 rotation parameters, 3 translation parameters, and 1 scale parameter in FLAME, we use two parameter settings of 60 parameters and 160 parameters to train our Trans3DHead and DAD-3DNet on the DAD-3DHeads dataset, respectively.
>
> As shown in Table 8 in the appendix, regressing more parameters is helpful for better reconstructing 3D head shape for both DAD-3DNet and the proposed Trans3DHead. Besides, the proposed Trans3DHead is superior to DAD-3DNet in all three parameter settings. When only regressing 60 parameters, our model shows more significant advantages over DAD-3DNet, which also indicates that the cross-attention mechanism in Trans3DHead is effective to achieve the information communication among different elements of the 3DMM parameters.
>
> ## [Question 4]
> An analysis of the computational cost of the method in comparison with other solutions is missing. For landmarks detection the capacity of the approach to work in real time is important. Authors should clarify this point. Authors should discuss and illustrate the computational cost of the method in comparison with other solutions is missing.
> ## [Answer]
> Thank you for the comment. Please refer to the Common Questions and Responses 2.

---

### Official Review · Reviewer_QjYA · 2023-11-01

**Soundness:** 2 fair
**Presentation:** 2 fair
**Contribution:** 3 good
**Rating:** 5
**Confidence:** 3

**Summary:**

This paper proposes a new method for 3D face landmarks detection. The contributions are:
- The paper also jointly estimate 2D face landmarks.
- The paper employs a DETR like approach and each estimated parameter is associated with a query embedding. This can help information communication in joint estimation of all the parameters.
- The paper also proposes a module to improve the model efficiency using low resolution features.
- The proposed method predicts residuals from the average face instead of directly predicting the original face.
- And Euler Angles loss is proposed to improve the performance on atypical head poses.

Experiments show the competitiveness of proposed method compared with baselines.

**Strengths:**

+ The presentation of the paper is good.
+ The use of query embeddings and cross attention can help information communication in joint estimation of all the parameters. And this is interesting and novel.
+ Visual demos show the effectiveness of the proposed method in 3D pose and landmarks estimation.

**Weaknesses:**

- For the qualitative comparisons, only DAD-3DNet is compared with. From table 5, DAD-3DNet is already worse than the proposed method. While SynergyNet, which performs better than the proposed method, is not compared with in the visual demos.
- section 3.2, the description about QA memory is not super clear. Does the proposed method use more features maps to trade for a lower resolution? What is the benefit in doing this?
- Due to large variation of head poses and face shape, it might not be a good idea to compute the average face and predict the residuals.
- The paper proposes a Euler Angles loss, but from figure 4, it requires an estimation of the Euler angles from predicted 3DMM parameters. Therefore another module needs to be introduced do the estimation. This module introduces additional errors and might not benefit the supervision for 3DMM parameters.
- From table 3, the QA module seems not to improve the accuracy compared with baseline.
- From table 5, the proposed method is not quantitatively better than SynergyNet 3DV 2021.

**Questions:**

- The paper first mentions "memory" in the second paragraph in section 3.1. But there is no context and explanation about it. What does memory mean? Which part does it correspond to in the network structure?
- Can the authors give more explanation and proof about the QA module and its effectiveness? Does the module use more feature maps to trade for lower resolution?
- From section 3.3, the paper uses multiple decoders. What are the decoders like? Why are multiple decoders used? Do they make it computationally less efficient? There are no explanation about the usage of multiple decoders. Maybe I miss something.
- section 3.4.1, how are Euler angles estimated from 3DMM parameters? Is a neural network used here?

---

> ### Author Response · Authors · 2023-11-22
> **Responses to Reviewer QjYA**
>
> ## [Question 1]
> For the qualitative comparisons, only DAD-3DNet is compared with. From table 5, DAD-3DNet is already worse than the proposed method. While SynergyNet, which performs better than the proposed method, is not compared with in the visual demos.
> ## [Answer]
> Thank you for the comment. Please refer to the Common Questions and Responses 1 for the comparison to SynergyNet.
>
> In the 3D head alignment task, the DAD-3DNet is the most relevant work and the closed competitor to the proposed Trans3DHead, so the most important experimental comparisons of this work are shown in Table 1. Thank you for the encouraging statement that “DAD-3DNet is already worse than the proposed method”.
>
> Anyway, including SynergyNet for visual comparison is indeed a good idea. Thank you for your suggestion. Besides the new Figure 10, to further illustrate the difference between the SynergyNet and the proposed Trans3DHead, we have additionally visualized some examples of head pose estimation predicted by the SynergyNet and the proposed Trans3DHead on the AFLW2000-3D dataset, as shown in Figure 11 in the appendix. It can be observed that the SynergyNet performs slightly better in predicting pitch and roll, while the proposed model has more advantages in handling side faces.
>
> ## [Question 2]
> section 3.2, the description about QA memory is not super clear. Does the proposed method use more features maps to trade for a lower resolution? What is the benefit in doing this? Can the authors give more explanation and proof about the QA module and its effectiveness? Does the module use more feature maps to trade for lower resolution?
> ## [Answer]
> Thank you for your comment. Using low-resolution feature maps often leads to a decrease in accuracy. Based on an observation, we find that in Transformers the queries will act differently if the memory is of higher resolutions. Therefore, QAMem is proposed to simulate the working mechanism of high-resolution feature maps in the actual low-resolution feature maps we used to improve the accuracy. In practice, the QAMem layer can be simply implemented as a convolutional layer with 1 $\times$ 1 kernel, 1 stride, and N groups. Note that we do not use more feature maps to trade for a lower resolution. In our implementation of QAMem in Eq. (3), the feature map channels are the same $d$ channels, but N groups of 1 $\times$ 1 convolutions are applied to transform the $d$ channels per query. As a result, QAMem provides discriminative values for different queries with marginal additional computation, and enhances discriminative abilities of the model on using low-resolution feature maps.
>
> Note also that the implementation of applying high-resolution feature maps usually uses additional network for up-sampling, such as the commonly used FPN structure where the parameters, computation, and memory requirement are significantly larger.
>
> ## [Question 3]
> Due to large variation of head poses and face shape, it might not be a good idea to compute the average face and predict the residuals.
> ## [Answer]
> The average face provides a general initial topology of the landmark detection with which the predicting task will be easier with small or not too large head poses. When encountering large head poses, the decoders with sufficient training are also able to output larger residual to correct the initial landmarks. Thank you for pointing out this. To address large variation of head poses and face shape better, a better way could be considering multiple average faces, which will be our future study.
>
> ## [Question 4]
> The paper proposes a Euler Angles loss, but from figure 4, it requires an estimation of the Euler angles from predicted 3DMM parameters. Therefore another module needs to be introduced do the estimation. This module introduces additional errors and might not benefit the supervision for 3DMM parameters. section 3.4.1, how are Euler angles estimated from 3DMM parameters? Is a neural network used here?
> ## [Answer]
> The estimation of the Euler angles from the predicted 3DMM parameters is a deterministic matrix calculation, with no additional network or learnable parameters in introducing errors. Specifically, the 3DMM parameters used in DAD-3DNet and the proposed Trans3DHead are adapted to FLAME, which are extensions of the standard 3DMM parameters. The number of the extended 3DMM parameters regressed in the final Trans3DHead is 413, including 403 3DMM shape parameters, 6 rotation parameters, 3 translation parameters, and 1 scale parameter. Then, the Euler angles are estimated by a deterministic matrix calculation from the predicted 6 rotation parameters, because any orientation can be expressed as a composition of rotations. In our implementation, we use SciPy, a scientific computation Python library, through scipy.spatial.transform.Rotation.as_euler().

---

> > ### Author Response · Authors · 2023-11-22
> > **Responses to Reviewer QjYA**
> >
> > ## [Question 5]
> > From table 3, the QA module seems not to improve the accuracy compared with baseline.
> > ## [Answer]
> > Thank you for your comment. The QAMem module is mainly proposed for the landmark detection subtask, so it has a relatively larger improvement on NME. For other metrics, although no improvements are observed, they are quite comparable. Note that the baseline is already a very good baseline of the proposed two-branch Transformer based method.
> >
> > ## [Question 6]
> > From table 5, the proposed method is not quantitatively better than SynergyNet 3DV 2021.
> > ## [Answer]
> > Thank you for the comment. Please refer to the Common Questions and Responses 1 for the comparison to SynergyNet.
> >
> > ## [Question 7]
> > The paper first mentions "memory" in the second paragraph in section 3.1. But there is no context and explanation about it. What does memory mean? Which part does it correspond to in the network structure?
> > ## [Answer]
> > Thank you for pointing out this. With the term “memory” we used the definition in DETR [r1]. The original definition of memory in DETR is the output of a Transformer encoder, which serves as the key and value of the cross-attention in the decoder. As analyzed in Table 3 of this paper, in the proposed Trans3DHead stacking more decoder layers improves the performance while more encoder layers have an inconspicuous effect or even reduce the results. Therefore, to balance the accuracy and efficiency, we remove the encoder layer and only use two decoder layers for the proposed Trans3DHead. Accordingly, the memory in our work is the final feature map extracted by the backbone, i.e., the feature output by a convolutional layer with 1 $\times$ 1 kernel and 512 channels after the ResNet-50 backbone.
> >
> > [r1] Nicolas Carion, Francisco Massa, Gabriel Synnaeve, Nicolas Usunier, Alexander Kirillov, and Sergey Zagoruyko. End-to-end object detection with transformers. In European Conference on Computer Vision, pp. 213–229, 2020.
> >
> > ## [Question 8]
> > From section 3.3, the paper uses multiple decoders. What are the decoders like? Why are multiple decoders used? Do they make it computationally less efficient? There are no explanation about the usage of multiple decoders. Maybe I miss something.
> > ## [Answer]
> > The structure of the decoder is shown in Figure 1, which is similar as in vanilla Transformers. Multiple decoders with the same structure can be used in a cascade as in vanilla Transformers. As shown in Figure 7 in the appendix, we conduct experimental analysis to verify that multi-layer decoder can fine-tune the prediction results layer by layer. Furthermore, the distribution of absolute values of normalized residual coordinates for each decoder layer is visualized in Figure 8. The visualization results further indicate that as the decoder layer deepens, more residual coordinates gradually approach 0, becoming sparse, which realizes refining the landmarks layer-by-layer and makes the prediction task easier. This strategy will bring some more computation, and thus two decoders are used in the final Trans3DHead to balance the computation and accuracy.

---

### Official Review · Reviewer_eTNq · 2023-11-01

**Soundness:** 2 fair
**Presentation:** 3 good
**Contribution:** 2 fair
**Rating:** 5
**Confidence:** 4

**Summary:**

The paper presents an efficient multi-task transformer for 3D face alignment, self-attention and corss-attention mechanisims were sused to enhance information communication among different elements of the network. Query aware memory (QAMem) is also designed to remove dependence on high-resolution feature maps. Experiments on two public benchmarks show that the approach can achieve resonable peformance.

**Strengths:**

The presentation is clear and easy to follow.
The experiments seem to be intensive and resonable results were achieved.

**Weaknesses:**

1) The contributions seem to be a combination of deep learning tricks like multi-task structure, QAMem module, MARR and Euler Angles Loss, the improvement of each module seems to be incremental and the combination of these incremental contributions, do not become a significant contribution.

2) The pose estimation results on AFLW2000-3D dataset, shown in Table 2, don't support that the proposed approach achieve better performance than SOTA. Th MAE, pitch and roll of the proposed approach are not as good as SynergyNet published in 2021.

3) The ablation study in Table 3 don't support the effeictiveness of proposed modules as well. For example, the performance of Cham Dist for Trans3DHead is not as good as the baseline.

I have read the response from authors and the comments of other reviewers. Most of the reviews also questioned the experiments and the results, which does not seem to be convincingly better than sota approaches, so I keep my orginal ratings.

**Questions:**

Further discussion of the novelty of the work need to be elaborated, I don't thnk the contribution listed in the paper, make a good work for top conference like ICLR.

---

> ### Author Response · Authors · 2023-11-22
> **Responses to Reviewer eTNq**
>
> ## [Question 1]
> The contributions seem to be a combination of deep learning tricks like multi-task structure, QAMem module, MARR and Euler Angles Loss, the improvement of each module seems to be incremental and the combination of these incremental contributions, do not become a significant contribution.
> ## [Answer]
> Thank you for your review but we do not agree. This work is well motivated to solve respective problems instead of using deep learning tricks to improve the performance. First, this is the first work in regressing 3DMM parameters through Transformers, which can help information communication among 3DMM parameters via attention mechanism in joint estimation. With this architecture, the proposed multi-task framework for 3D head alignment simultaneously predicts 3D head shape and 2D landmarks. This not only facilitates applications by directly providing multiple available outputs, but also helps mutual improvements between the two branches. Our experimental findings show that with this design the proposed method is more robust with reducing 3DMM parameters (see A.1.6 and Table 8 in appendix), thanks to the unified Transformer architecture for two branched tasks.
>
> Second, the proposed lightweight QAMem module is an interesting and valuable idea (supported by Reviewers QjYA and PGZe) for dense prediction tasks requiring high-resolution feature maps. It ingeniously uses separate memories with low-resolution feature maps and designs an efficient implementation to get rid of the high-resolution feature map requirement. The QAMem design enables improved accuracy with minimal additional computational burden.
>
> These are the two major innovations in this paper. We do not agree that they can be simply categorized into deep learning tricks or incremental combinations.
>
> ## [Question 2]
> The pose estimation results on AFLW2000-3D dataset, shown in Table 2, don't support that the proposed approach achieve better performance than SOTA. Th MAE, pitch and roll of the proposed approach are not as good as SynergyNet published in 2021.
> ## [Answer]
> We may did not make it clear. Thank you for the comment. Please refer to the Common Questions and Responses 1 for the comparison to SynergyNet.
>
>
> ## [Question 3]
> The ablation study in Table 3 don't support the effeictiveness of proposed modules as well. For example, the performance of Cham Dist for Trans3DHead is not as good as the baseline.
> ## [Answer]
> Thank you for the comment. The measurements and comparisons in Table 3 are comprehensive, including evaluations of not only 3D face shape prediction (Chamfer distance), but also face landmark detection (NME), head pose estimation (pose error), and 3D head shape estimation (Z5). In DAD-3DHeads benchmark, Chamfer distance is used for measuring the 3D shape of human face, and the proposed method achieves quite comparable results with the baseline in terms of Chamfer distance (2.796 vs. 2.791). Except this the proposed method outperforms the baseline in all other metrics. Especially, the proposed model performs better than the baseline with Z5 (0.9585 vs. 0.9578). As Z5 is used for measuring 3D head shape, it is more important than Chamfer distance for 3D face shape only.

---

### Author Response · Authors · 2023-11-22
**Common Questions and Responses**

# [Common Questions and Responses]

We thank all the reviewers for their valuable feedback. There are some concerns commonly raised by two or more reviewers. We give general responses here to address the common concerns.

## [Question 1]

The comparison between SynergyNet and the proposed Trans3DHead ([Question 2] by reviewer eTNq, [Question 1] and [Question 6] by reviewer QjYA, and [Question 2] by reviewer zzMV).

## [Answer 1]

It’s worth noting that this work is proposed for the full 3D head alignment task, in contrast to the traditional 3D face alignment task addressed by SynergyNet. With the differentiable FLAME model which integrates 3DMM parameters, the proposed method is able to reconstruct the full 3D head shape including neck, while SynergyNet can only reconstruct face shape. 3D pose estimation is only a side product of the proposed method, but not the main focus. However, as there are not many existing works working on full 3D head alignment (see Table 1), we provide additional comparison in Table 2 for 3D pose estimation so that many existing traditional methods can be compared. Please note that, though not better, the proposed Trans3DHead achieves almost the same MAE (3.38 vs. 3.35) as that of SynergyNet on the AFLW2000-3D dataset in the head pose estimation task. This is already much better than DAD-3DNet (3.66) which is our closed competitor. Figure 11 in the appendix additionally visualizes the results.

To further illustrate the difference between SynergyNet and the proposed Trans3DHead, we have additionally included the 3D face shape predicted by SynergyNet to Figure 10 in the appendix. It can be clearly observed that the proposed method not only obtains more accurate shapes than SynergyNet but also reconstructs the entire 3D head.


## [Question 2]

Efficiency of the proposed Trans3DHead ([Question 4] by reviewer zzMV, and [Question 2] by reviewer PGZe).

## [Answer 2]

We have conducted experiments to estimate the average inference speed of the DAD-3DNet and the proposed Trans3DHead on the same platform (1 NVIDIA V100 GPU). With a total of 2746 testing images, the Frame Per Second (FPS) of the DAD-3DNet is 112, while that of the proposed Trans3DHead is 115, slightly better than DAD-3DNet. They are both quite efficient. While we design QAMem in Trans3DHead to avoid using high-resolution heatmaps in a Feature Pyramid Network (FPN) architecture in DAD-3DNet, and hence efficiency is improved, the Transformer head branches in our design, though improves the accuracy, is not that efficient than simple FC layers used in DAD-3DNet. Therefore, in overall Trans3DHead is only slightly more efficient than DAD-3DNet.

---

### Meta-Review · Area_Chair_6KNQ · 2023-12-13

**Metareview:**

The paper proposes an efficient multi-task transformer for 3d face landmark detection and 3d face alignment. The network uses self-attention and cross-attention mechanisms, query aware memory and Euler Angles loss to enhance the results.
Strengths: the proposed method has some novelty.
Weaknesses: (1) the technical contribution is limited, which is a combination of deep learning tricks, (2) the experimental results are not convincing (quantitative results only for one dataset, performance comparable but in some case below existing methods).
The paper should include more convincing results and in-depth description of the proposed techniques.

**Justification For Why Not Higher Score:**

The paper got all negative feedbacks from four reviewers who are the experts in this domain.

**Justification For Why Not Lower Score:**

N/A

---

### Decision · Program_Chairs · 2024-01-16

Reject